# Revealing Insights: A Comprehensive Overview of Gastric Outlet Obstruction Management, with Special Emphasis on EUS-Guided Gastroenterostomy

**DOI:** 10.3390/medsci12010009

**Published:** 2024-02-01

**Authors:** Dimitrios Ziogas, Thomas Vasilakis, Christina Kapizioni, Eleni Koukoulioti, Georgios Tziatzios, Paraskevas Gkolfakis, Antonio Facciorusso, Ioannis S. Papanikolaou

**Affiliations:** 11st Department of Internal Medicine, 251 Hellenic Air Force & VA General Hospital, 3 Kanellopoulou str., 11525 Athens, Greece; 2Hepatology and Gastroenterology Clinic, Charité Campus Mitte, Charitéplatz 1, 10117 Berlin, Germany; thomas.vasilakis@charite.de; 3Hepatogastroenterology Unit, Second Department of Internal Medicine-Propaedeutic, Medical School, National and Kapodistrian University of Athens, 12462 Athens, Greece; xristinakapiz@hotmail.com (C.K.); e.koukoulioti@gmail.com (E.K.); ispapn@hotmail.com (I.S.P.); 4Department of Gastroenterology, “Konstantopoulio-Patision” General Hospital, 3-5, Theodorou Konstantopoulou Street, Nea Ionia, 14233 Athens, Greece; g_tziatzios@yahoo.gr (G.T.); pgolfakis@gmail.com (P.G.); 5Department of Medical Sciences, University of Foggia, Section of Gastroenterology, 71122 Foggia, Italy; antonio.facciorusso@virgilio.it

**Keywords:** endoscopic ultrasound-guided gastroenterostomy, gastric outlet obstruction, endosonography, self-expandable metal stent, gastroenterostomy

## Abstract

Gastric outlet obstruction (GOO) poses a common and challenging clinical scenario, characterized by mechanical blockage in the pylorus, distal stomach, or duodenum, resulting in symptoms such as nausea, vomiting, abdominal pain, and early satiety. Its diverse etiology encompasses both benign and malignant disorders. The spectrum of current treatment modalities extends from conservative approaches to more invasive interventions, incorporating procedures like surgical gastroenterostomy (SGE), self-expandable metallic stents (SEMSs) placement, and the advanced technique of endoscopic ultrasound-guided gastroenterostomy (EUS-GE). While surgery is favored for longer life expectancy, stents are preferred in malignant gastric outlet stenosis. The novel EUS-GE technique, employing a lumen-apposing self-expandable metal stent (LAMS), combines the immediate efficacy of stents with the enduring benefits of gastroenterostomy. Despite its promising outcomes, EUS-GE is a technically demanding procedure requiring specialized expertise and facilities.

## 1. Introduction

Gastric outlet obstruction (GOO) is the intrinsic or extrinsic mechanical blockage of gastric emptying at the level of the antrum or duodenum resulting from various benign or malignant conditions [1,2,3]. Most common benign conditions are gastric and duodenal ulcers. Other benign conditions causing GOOs are post ulcer stenosis, acute pancreatitis and pancreatic fluid collections, radiation-induced stricture, caustic ingestion, foreign body, benign tumor (adenoma, lipoma), Crohn’s disease, and eosinophilic gastroenteritis [1,2,4]. In the past, ulcer-induced GOO was very common (up to 90% in 1990), but with the advent of PPIs and helicobacter pylori eradication, the most common cause of GOO nowadays is malignancy (50–80%) [1]. Malignant gastric outlet obstruction (malignant GOO) can result from cancers arising from the stomach, duodenum, pancreas, biliary tree, and liver, as well as from metastases [1,4,5]. In the Western world, malignant GOO is most frequently attributed to pancreatic adenocarcinoma, while gastric adenocarcinoma is responsible for most malignant GOOs in Asia [5].

Individuals suffering from GOO typically experience epigastric pain, early satiety, nausea, and vomiting. In cases of malignant GOO, these symptoms can also be caused from the primary tumor or appear as a side effect of radiation or chemotherapy. [4] For this reason, a careful differential diagnosis should be performed. Diagnosis should be made early since malnutrition can appear as a result of poor oral food intake, which in turn can negatively influence patient prognosis [1,4].

Clinical examination may reveal a palpable mass due to stomach distension (15% of the cases) [1]. This can be easily confirmed initially through transabdominal ultrasound or an X-ray of the abdomen, although a CT scan will usually follow. Both exams can show the enlarged stomach and the lack of small or large intestine distension. Thus, intestinal blockage can be excluded, which is also an important differential diagnosis of vomiting. Furthermore, it is very important to assess the presence of more than one stenotic segments or ascites (benign or malignant) [2]. Contrast studies utilizing barium or water-soluble contrast agents prove to be valuable diagnostic tools, particularly in patients with a documented history of malignancy who present with a sudden onset of vomiting. The absence of contrast passage in the small intestine indicates a complete GOO. A CT scan of the abdomen offers the advantage of revealing the extent of the disease, especially in the case of malignancy, the presence of metastasis or peritoneal carcinomatosis. Upper gastrointestinal endoscopy can directly determine the cause of the stenosis with or without biopsy and in cases of extraluminal compression, endoscopic ultrasonography is needed for tissue sampling as well as locoregional staging. All the above information along with estimated patient survival and patient preferences will determine which is the most appropriate therapeutic management [4,5].

In previous decades, surgical gastroenterostomy (SGE) has been the treatment of choice for GOOs. Nowadays, self-expandable metallic stents (SEMSs) and endoscopic ultrasound-guided gastroenterostomy (EUS-GE) have gained wide acceptance and have replaced surgery for selected patients [2,4].

In this narrative review, we describe the available methods for the management of GOOs, explain patient selection for each method, and compare available methods based on current data discussing their efficacy and safety. Notably, our primary emphasis is on EUS-GE, with special attention given to the occurrence of stent misdeployment during the procedure.

## 2. Management

### 2.1. Surgical Gastrojejunostomy

This method was considered the traditional bypass method to alleviate obstructive symptoms of GOO and ameliorate patients’ quality of life before the emergence of less invasive endoscopic techniques. It can be performed either as an open or a laparoscopic surgery. It is important that the gastrostomy is formed 3–5 cm proximal to either the obstruction or the pylorus. The anastomosis needs to be placed at the distal stomach on the greater curvature, thus avoiding a too high placement, as this could result in biliary reflux and delayed gastric emptying. A jejunal loop 10–15 cm distal to the ligament of Treitz is chosen and placed close to the stomach in an isoperistaltic conformation and not under tension either in an antecolic or a retrocolic fashion [6].

Common complications of the SGE include infections, gastroparesis, hemorrhage, anastomotic leak (mostly between the third to fifth postoperative day), and marginal ulcer (gastrojejunostomy predisposes the part of the jejunum nearest to the stomach to ulcer development because it lacks the protective mechanisms of the duodenum) [6,7]. Laparoscopic SGE demonstrates improved morbidity and mortality rates, as well as shorter hospitalization, quicker resumption of eating, diminished intraoperative bleeding, and decreased requirements for opiates postoperatively [8].

Surgery is performed on patients with life expectancy of at least several months, a duration deemed ample to navigate and overcome the risk and complications of the surgical intervention [4]. Also, surgery may be performed as a salvage option in selected patients after unsuccessful EUS-GE [9].

### 2.2. Self-Expandable Metallic Stents

SEMS is the oldest one of the endoscopic techniques used for GOO. SEMSs are available in a condensed form within a device designed to be inserted through the working channel of an endoscope, over a wire that has previously been inserted through the obstruction [4] The stent can then be deployed with or without the use of fluoroscopy. The stent automatically expands to reach its maximum diameter within the following 24 to 48 h. An endoscope with a broad working channel (i.e., =3.7 mm) is required. The majority of cases are managed using therapeutic gastroscopes; however, situations involving enlarged stomachs or strictures in the distal duodenum may be more effectively addressed with a colonoscope or, in some cases, a duodenoscope [10,11].

There are three types of SEMS: uncovered (uSEMS), partially covered (pcSEMS), and fully covered (fcSEMS). Generally, it appears that both uSEMS and fcSEMS exhibit comparable effectiveness. Studies suggest that all SEMSs have similar technical success (ranging between 89% and 98%) and clinical success, typically characterized as the alleviation of obstructive symptoms and increase in oral consumption [5], ranging between 63% and 93% [12]. The choice of the stent type is based on the individual characteristics of the stenosis, knowing that fcSEMS have higher migration rates, whereas stent ingrowth rate is higher with uSEMS. The latter offer the additional advantage of improved bile outflow (through the stent’s mesh interstices) when the stent is positioned across the papilla in the duodenum [4,5].

The rate of adverse events (AEs) associated with SEMS placement varies between 0% and 30%, based on the definition applied in each study. This may include minor AEs such as nausea, vomiting, and mild abdominal pain, or major AEs, such as bleeding, perforation, stent migration/displacement, or cholangitis [5]. Delayed AEs are commonly associated with stent dysfunction, resulting from migration or blockage due to food impaction and/or tumor ingrowth/overgrowth. Several methods (e.g., stent clipping or suturing, anti-migratory design) are recommended to diminish the risk of stent migration [13].

The placement of a SEMS is the first-line strategy in patients with short life expectancies (<3 months) and can also be an alternative, if EUS-GE is not successful [9].

### 2.3. EUS-Guided Gastroenterostomy

EUS-GE is a new endoscopic treatment for GOO offering equivalent efficacy compared to other techniques but with possibly fewer adverse events in experienced hands. During EUS-GE, the stomach and small intestine are fused with the application of a new type of stent, the lumen apposing metallic stent (LAMS), which was introduced in 2012 [14]. LAMS is a double flanged, fully covered stent that provides a stable anastomosis between two adjacent organs (Figure 1). The presence of broad flanges on both ends results in a better anchoring and allows for equal distribution of pressure on the luminal wall, thus decreasing the risk of migration [14]. For this procedure, a linear echoendoscope with forward or oblique viewing is needed [2]. LAMS can be deployed using many methods. They can be divided into two groups, the direct method and the assisted methods: a. the antegrade EUS-GE traditional downstream method, b. the antegrade EUS-GE rendezvous method, c. the retrograde EUS-enterogastrostomy and the EUS balloon occluded GE bypass (EPASS) [2,15]. The first four are the most frequently used [2,9]. Here, we describe the basic steps of each procedure [2,15,16]. The selection of the target site depends on the proximity of the specific segment of the small bowel to the gastric wall, as well as the presence of tumor involvement in the third part of the duodenum. When faced with widespread malignant infiltration in the stomach, performing a gastroenterostomy, whether through surgical or endoscopic means, may not be a viable option [2].

Antegrade EUS-GE direct method
**Step 1:** Visualize the target intestinal limb by injecting saline or diluted contrast (methylene blue) distal to the obstruction through an orojejunal tube that was previously inserted under endoscopic guidance.**Step 2:** Perform an EUS-guided puncture of the target intestinal limb using a 19-gauge needle.**Step 3:** Aspirate; methylene blue aspiration confirms correct localization of the needle in the target jejunal limb.**Step 4:** Pass either a guidewire through the needle, in order to place a LAMS over it, or use directly a cautery-enhanced LAMS (HOT AXIOS; Boston Scientific Corp.).
Antegrade EUS-GE direct method using the wireless endoscopic simplified technique (WEST)
**Step 1:** As in Technique 1.**Step 2:** Advance and deploy directly a cautery-enhanced LAMS (HOT AXIOS; Boston Scientific Corp.) in cut mode.Antegrade EUS-GE traditional downstream method
**Step 1:** Position a guidewire in the jejunal lumen past the obstruction under endoscopic guidance. Withdraw the endoscope and keep guidewire in place.**Step 2:** Using fluoroscopy advance a dilating balloon over the wire to the jejunum. Dilate the balloon.**Step 3:** From the stomach perform an EUS-guided puncture of the balloon with a 19-gauge needle.**Step 4:** Pass another guidewire through this needle into the jejunum.**Step 5:** Deploy the LAMS over the second guidewire.Antegrade EUS-GE with direct technique over a guidewire (DTOG)

This technique is applied, if the obstruction cannot be crossed.

**Step 1:** Administer intravenous anticholinergic agent to slow bowel movements**Step 2:** Puncture the target intestinal limb with a 19-gauge needle.**Step 3:** Fill the jejunal limb with contrast medium through the needle and insert a guidewire using fluoroscopy.**Step 4:** Advance the cautery-enhanced LAMS catheter over the guidewire into the jejunal limb in cut mode. Deploy the LAMS.

Less common techniques include the following:5.Antegrade EUS-GE rendezvous method

**Steps 1 to 3:** See Technique 3.**Step 4:** Entrap the puncturing guidewire in the dilating balloon that was punctured, or capture it with an ERCP extraction balloon and/or basket and pull it back outside the mouth, in order to secure it.**Step 5:** Deploy the LAMS using this guidewire under traction.

6.Retrograde EUS-EG Enterogastrostomy

**Steps 1 to 4:** See Technique 5.**Step 5:** Advance a therapeutic endoscope over the guidewire till you locate the inserted guidewire in the duodenum/jejunum.**Step 6:** Deploy the LAMS in a retrograde fashion by opening the gastric flange first.

7.EUS balloon occluded GE Bypass (EPASS)

**Step 1:** Using a double-balloon enteroscope (DBE) position a guidewire in the jejunum.**Step 2:** Withdraw the DBE, while keeping the overtube in the antrum or duodenal bulb.**Step 3:** Use a double-balloon-occlusion catheter. This catheter has two balloons (with 20 cm distance between them). Insert it distal to the obstruction under endoscopic control and then inflate both balloons to stabilize the target intestinal limb. Then fill this segment with contrast.**Step 4:** Perform an EUS-guided puncture between the two balloons.**Step 5:** Deploy the LAMS.

Regarding complications of EUS-GE, the existing literature shows that EUS-GE is effective and safe. The first study assessing EUS-GE was a retrospective study by Khashab et al. [17], in which EUS-GE was performed in 10 patients with GOO (3 malignant and 7 benign) through the use of the direct or the balloon-assisted technique. EUS-GE had a technical success of 90% and clinical success of 100%, while no adverse effects were described. According to a recent review including fourteen studies, technical success of EUS-GE ranged from 87 to 100% and had a clinical success of 84–100%, irrespective of the technique performed [18,19,20,21]. A recent meta-analysis including twelve, mostly retrospective, studies (*n* = 285), conducted by Iqbal et al. [19], assessed the performance of EUS-GE in the treatment of GOO. Pooled technical success rates were 92% (95% CI: 88–95%) and clinical success rates were 90% (95% CI: 85–94%).

The rate of adverse effects ranges from 0 to 21%, encompassing stent misdeployment, pneumoperitoneum, gastric leak, bleeding, peritonitis, or abdominal pain [22,23]. Itoi et al. reported only a 10% stent misdeployment, but no additional AEs were documented, and there were no instances of stent migration or occlusion requiring further interventions when applying the EPASS method [24]. Mahagis et al. described an acute exacerbation of gastric outlet obstruction after an EUS-GE. The cause was that the LAMS axis was set in a distinctly angled orientation towards the afferent limb, leading to the formation of a closed loop. This was managed by placing a fcSEMS through the LAMS into the efferent loop [25]. The fluctuation of the technical success rates but also of the AE rates may be attributed to its technical difficulty and long learning curve.

Two studies also examined EUS-GE regarding long-term efficacy and patency. In a retrospective study of 57 patients by Kerdsirichairat et al., clinical success was achieved in 89.5% of patients over a median follow-up period of 196 days for patients with malignant GOO and 319.5 days for patients with benign GOO [26]. During the follow-up time, reintervention with upper endoscopy due to recurrence of GOO symptoms was needed in eight patients (15.1%), of whom stent occlusion was present in two. Similarly, On et al. reported that across a median follow-up period lasting 162 days, reintervention for recurrent GOO was required for only one patient from the 21 with a technically successful EUS-GE and available follow-up data [27].

## 3. Comparison of the Available Treatments

### 3.1. Comparison of the Available LAMS

Commonly used types of LAMS include the AXIOS stent (Boston Scientific Corp., Marlborough, MA, USA), which is the type of stent initially used, Niti-S Spaxus stent (Taewoong Medical Co., Gimpo, Republic of Korea), and NAGI stent (Taewoong Medical Co., Goyang, Republic of Korea). LAMS deployment initially requires the guidewire insertion and then the expansion of the transluminal tract with a dilation balloon. More recently, electrocautery-enhanced LAMS (EC-LAMS, HOT AXIOS) have been created: permitting incision, tract expansion, and stent deployment in a one-step procedure without the guidewire utilization, thus simplifying the procedure and reducing the risk of complications. Moreover, by avoiding the guidewire implementation, the risk of small bowel moving away from the stomach diminishes [28], thus reducing the risk of misdeployment of the distal flange.

Historically, the 15 mm LAMS has been most commonly used for EUS-GE, but recently 20 mm LAMS has been developed as an alternative choice. It is hypothesized that the wider lumen of 20 mm LAMS may be associated with better patency and improved clinical outcomes, as its lumen size is not divergent from the physiologic gastric outlet size (20 to 23 mm) [29]. However, this theoretical advantage may be counterbalanced from increased risk of adverse effects, due to the more difficult deployment of the large flange size. In a multicenter retrospective study including 267 patients, Bejjani et al. [30] reported that 20 mm LAMS was comparable to the 15 mm LAMS concerning effectiveness and safety for managing malignant GOO. Notably, more individuals in the 20 mm LAMS group endured a mushy/nutrient-rich diet. This study along with twelve more, were pooled in a recent meta-analysis evaluating the efficiency and safety of 20 mm LAMS in comparison to 15 mm LAMS for the treatment of GOO [31]. No difference was found in the pooled technical success rate (20 mm: 92.1% vs. 15 mm: 93.2%, *p* = 0.47), pooled clinical success rate (89.6% vs. 88.6%, *p* = 0.62), and pooled adverse event rate (14.7% vs. 11.4%, *p*= 0.08) between the two groups. However, a noteworthy disparity was noticed in the pooled reintervention incidence, which was significantly decreased for 20 mm LAMS (3.5% vs. 10.3%, *p* = 0.008). Many variables should be considered regarding the stent diameter that should be used in each patient, such as the echoendoscopist’s experience, the selected EUS-GE technique, the etiology of GOO, and the anatomy of upper gastrointestinal tract. More data are required to draw more reliable conclusions.

### 3.2. Comparison for the Different Techniques for EUS-GE

Few data are available concerning the comparison of the available EUS-GE techniques in regard of efficiency and safety. Chen et al. performed a retrospective multicenter study including 74 patients to assess and contrast the direct with the balloon-assisted method [32]. Technical success, clinical efficacy, and incidence of AEs exhibited no significant differences between the two methods, while the direct technique was associated with an average of 54.5 min shorter procedure duration than the balloon method. Recently, a meta-analysis including 20 studies demonstrated a significant lower AEs rate for the direct technique compared to balloon-assisted techniques, either single-balloon or double balloon (EPASS) (9.3% vs. 21.4%, *p* = 0.001) [33]. The incidence of severe AEs was also decreased for the direct technique, although without significant difference among the two groups (3.1% vs. 8.2% *p* = 0.099). This comes in contrary to the hypothesis that the direct technique, as an unassisted method, would have an greater risk of AEs in comparison with the balloon-assisted methods [32,33]. Technical and clinical success rates were also similar between the two procedures. In subgroup analysis, no differences were found between the single-balloon or double-balloon methods regarding technical success, clinical success, or procedure time and only the total AEs rate was significantly lower for the single balloon group (8.9% vs. 28.5%, *p* = 0.004).

### 3.3. Comparison between Methods

#### 3.3.1. EUS-GE vs. SGE

So far, only a few studies exist that assess the performance of EUS-GE compared to SGE in malignant GOO. A recent meta-analysis by Bomman et al., pooling six retrospective studies and 484 patients, demonstrated that EUS-GE showed a markedly reduced rate of technical success (95% CI: 0.054–0.702; *p* = 0.012; Q = 1.909; I2 = 0) but had a favorable clinical success rate compared to SGE; however, it was without statistical significance (OR = 1.566; 95% CI: 0.585–4.197; *p* = 0.372) [34]. Moreover, EUS-GE was related with a lower rate of adverse effects (OR = 0.295; 95% CI: 0.172–0.506; *p* < 0.005; Q = 0.40: I2 = 0). This result was expected given the minimally invasive nature of EUS-GE in contrast to a surgical or laparoscopic intervention such as SGE. The inferiority of EUS-GE in technical success in the aforementioned meta-analysis could be attributed to the fact that as a new intervention it is still technically challenging and demands further expertise. Of note, two more recent studies by Jaruvongvanich et al. and Canakis et al. documented comparable technical success for EUS-GE vs. SGE in the alleviation of both malignant and benign GOO (98.3% vs. 100% and 97.9% vs. 100%, respectively) [35,36]. In their retrospective study, Jaruvongvanich et al. also showed that in a median observation duration of 185.5 days, EUS-GE in comparison with SGE was related with significantly increased clinical success rate (98.3% vs. 90.4%, *p* = 0.002), significantly lower reintervention rate for recurrent GOO (0.9%, and 13.7%, respectively, *p* < 0.0001) and lower adverse effects rate, indicating that EUS-GE could secure long-term patency and clinical improvement equally to SGE [36]. Furthermore, EUS-GE compared to SGE required a shorter post-procedure length of hospital stay (MD: –5.95; 95% CI: –6.99 to –4.91; *p*  <  0.001; I2  =  95%) and led to faster return to both oral consumption and chemotherapy in cases of mGOO [35,37]. This is of great importance since these patients often have unresectable tumors and palliative chemotherapy is the only available therapy to improve their prognosis. Altogether, it is apparent that when performed in specialized and experienced centers, EUS-GE is an excellent alternative to SGE as a treatment of GOO, which however needs further standardization and expertise. Thus, more studies, ideally randomized controlled trials comparing these two interventions are essential to reach a more reliable conclusion.

#### 3.3.2. EUS-GE vs. SEMS

A recent meta-analysis consisting of eight studies demonstrated that EUS-GE had significantly higher clinical success (OR, 5.08; 95% CI, 3.42–7.55) and comparable technical success (OR, 0.44; 95% CI, 0.18–1.12) for the management of malignant GOO compared to SEMS [38]. Regarding the adverse effects rate, the authors found no statistically significant difference (OR, 0.57; 95% CI, 0.29–1.14). In addition, another meta-analysis with five retrospective studies and 659 patients showed that the rates of both overall and severe (per ASGE Lexicon) adverse effects were similar between EUS-GE and SEMS [39]. This fact is remarkable, given that as a more invasive procedure, EUS-GE would be expected to have more severe adverse effects. Van Wanrooij et al. [40] performed a multicenter retrospective study involving 214 patients (107 received EUS-GE and 107 SEMS), 176 of which were compared, using propensity score matching. Clinical success of EUS-GE vs. SEMS was higher (91% vs. 75%; *p* = 0.008) and the rate of stent dysfunction was decreased for EUS-GE (1% vs. 26%; *p* < 0.001). EUS-GE demonstrated a shorter median time for achieving clinical success (1 vs. 2 days; *p* < 0.001), while technical success and the median duration of hospitalization were comparable in both groups. Similarly, a recent prospective study conducted at a single center investigated the efficiency of EUS-GE via the WEST method vs. SEMS with 28 patients undergoing EUS-GE and 28 matched patients undergoing SEMS. The EUS-GE group demonstrated superior clinical success (100% vs. 75.0%, *p* = 0.006) and a marked decrease in symptom relapse throughout the observation period (3.7% vs. 33.3%, *p* = 0.02). Furthermore, EUS-GE was associated with a reduced time to clinical success vs. SEMS (2 vs. 3 days, *p* = 0.03) [41]. Teoh AYB et al. reported the first multicenter, randomized, controlled trial assessing EUS-GE, by using the EPASS technique (*n* = 48), vs. SEMS (*n* = 49) for the treatment of malignant GOO [42]. Regarding the primary outcome of reintervention rate at 6 months, EUS-GE showed notable superiority, with only two patients (4%) requiring reintervention compared to 14 patients (29%) in the SEMS group (*p* = 0.0020; RR 0.15 [95% CI 0.04–0.61]). EUS-GE was also linked with longer stent patency and improved eating function at 1 month, as reflected by a significantly higher mean 1-month gastrointestinal quality of life index (GOOS) of 2.41 (SD 0.7) compared to the duodenal stent group with a mean of 1.9 (0.9; *p* = 0.012, r = 0.26). Technical success, clinical success, and adverse effects were comparable between the two interventions. The greater clinical efficacy without the need of reintervention for recurrent GOO, which is observed with EUS-GE compared to SEMS could be explained by the lower incidence of either migration or obstruction of LAMS compared to SEMS. In fact, stent obstruction due to tumor ingrowth occurs in as many as 30% of patients with uSEMS, since SEMS is often placed through neoplastic tissue. EUS-GE bypasses the obstruction site, offering better patency and more durable treatment. Moreover, LAMS are shorter than SEMS (1.5 cm vs. 6–12 cm, respectively) making food passage through them easier [40]. In summary, EUS-GE is more effective and identically safe to SEMS, with a lower rate of reintervention.

## 4. Technique Choice

The advantages and drawbacks of the procedures utilized in the management of GOO are summarized in the Table 1. Current guidelines of the European Society of Gastrointestinal Endoscopy suggest EUS-GE, when executed by experts, as an alternative to SEMS or SGE for the treatment of malignant GOO. For benign GOO, EUS-GE is suggested for individuals deemed unsuitable for surgery, with a weak recommendation supported by low-quality evidence [43]. The decision about which treatment to perform is complex and should be individualized, considering patient characteristics, the location and type of obstruction, the presence of ascites, and the experience of the endoscopic team. When adequate expertise is available, EUS-GE should be considered as a reliable choice equivalent to SGE for the treatment of malignant GOO in patients with expected life durations greater than three months. EUS-GE combined with EUS-guided biliary drainage (EUS-BD) is also feasible and efficient when malignant GOO co-exist with biliary obstruction. Double endoscopic bypass offers the benefit of prolonged stent viability compared to the traditional methods of SEMS and endoscopic retrograde cholangiopancreatography (ERCP), since the site of anastomosis is separate from the obstruction site [44,45]. On the contrary, SEMS could be the preferable option in case of an advanced disease leading to short-life expectancy of less than two to three months, and when EUS-GE is not feasible, for example in the presence of a large volume ascites, malignant ascites, or a diffuse infiltration of the gastric wall. Finally, and above all, randomized trials evaluating the performance of EUS-GE in comparison with the other two methods are required to better determine its role as a treatment option for GOO.

It should be pointed out that in cases where the above-mentioned therapies are contraindicated or have failed, a percutaneous gastrostomy for relieving gastric pressure, followed by the subsequent insertion of a jejunal feeding tube can provide palliative treatment.

## 5. Misdeployment

One of the most common difficulties that endoscopists will encounter during EUS-GE is stent misdeployment (SM). Collectively, the rate of SM ranges from 6.8% to 27%, even in expert hands [9]. For this reason, advanced endoscopists declare this method as a technically challenging procedure. Furthermore, misdeployment associated adverse events have hindered its dissemination thus far [9].

Currently, there is only one multicentric retrospective study that examines this issue and ways to manage it [9]. EUS-GE was performed with either the direct method or the antegrade EUS-GE traditional downstream method in 467 patients. The SM rate of this study was 9.85%. Interestingly, 73.2% of SM happened during the first 13 cases with only 17% occurring after 25 procedures. For this reason, ESGE recommends that this procedure takes place only in expert centers [46].

This study observed and classified misdeployment into the following four types: Type I describes the deployment of the distal flange in the peritoneum with no puncture of the small intestine (rate: 63.1%). Type II happens when the distal flange is deployed in the peritoneum, but puncture of small intestine was achieved (rate: 30.4%). Type III describes the deployment of the proximal flange in the peritoneum, which was observed only once, and Type IV happens when the distal flange is deployed in the colon, thus creating a gastrocolic anastomosis, which occurred twice. Overall, the adverse events that resulted from SM were graded as mild in 28 (60.9%) patients, moderate in 11 (23.9%) patients, severe in 6 (13%) patients, and fatal in 1 (2.2%) patient. Surgical treatment was necessary for 5 (10.9%) patients [9]. The following salvage strategies were implemented to correct SM.For SM Type 1 removal of the LAMS and closure the gastrotomy was performed using over-the-scope clips (OTSCs), through-the-scope clips (TTSCs), or endoscopic suturing. Two patients, who were treated conservatively with no closure of the gastrotomy, recovered with no adverse events. This type of error was addressed during the same endoscopic session via EUS-GE at the same or different gastric site or via placement of a duodenal stent. In this cohort, three patients were operated due to peritonitis and a SGE was performed [9]. Overall, the majority of Type I SM events were assessed as mild (*n* = 22, 75.9%), two were moderate (6.9%), and five severe (17.2%). The five severe cases included the three patients who required surgical treatment and two patients who were admitted to the ICU [9].For Type II SM, LAMS was removed and a new LAMS using the same EUS-GE method or the NOTES (natural orifice transluminal endoscopic surgery) method was placed; alternatively a fcSEMS through the initial misdeployed LAMS was placed to bridge the gap. Endoscopic closure of the gastrotomy only with OTSC or TTSC was also an alternative for five patients. Of those patients, two developed abdominal pain requiring narcotics and one patient pneumoperitoneum requiring drainage, but two experienced no adverse events. Only 50% of Type II SM patients experienced adverse events; abdominal pain requiring narcotics was the most common (28.6%). Overall, most Type II SM adverse events were graded as mild (*n* = 6, 42.9%) or moderate (*n* = 7; 50.0%). Only one patient was operated due to peritonitis [9].For the one case of Type III SM, the NOTES method was used, but it was not successful. For this reason, the LAMS had to be removed surgically and a SGE was performed [9].Type IV SM occurred twice; one was recognized during the procedure, while the second one was identified 3 weeks post procedure due to diarrhea induced by food intake. In both cases, LAMS was removed followed by endoscopic closure of the gastrocolic anastomosis, either with an endoscopic suturing or TTS [9].


An additional alternative salvage treatment in cases of SM was proposed by Fabbri et al. The authors performed a laparoscopic-assisted EUS-guided gastroenterostomy [47].

To acquire a proper understanding of the procedure and effectively mitigate the risks of complications, including stent misdeployment, novice echoendoscopists may consider following these expert-recommended tips: study the available videos of EUS-GE, discuss cases with experienced colleagues, and plan ahead and be prepared for a case of stent misdeployment. Additionally, the first procedures should be performed in the operating room with surgical support [48].

## 6. Conclusions and Future Perspectives

Irrespective of the etiology (malignant or benign), an efficient, durable, and safe intervention to alleviate GOO is essential in order to improve patient quality of life and prognosis. It is now apparent that in the hands of experts, EUS-GE is effective for the treatment of GOO and safer than the previous therapies, SEMS and SGE. This procedure integrates the advantages of the other two methods, providing long-term patency and efficacy without need of reinterventions of SGE, with a favorable safety profile such as SEMS. Nonetheless, most reports of EUS-GE arise from experienced centers, where this procedure is mainly used. EUS-GE is still technically challenging for many endoscopists and must be simplified and standardized further to be more commonly applied in clinical practice [49].

Therefore, in order for this method to become widely adopted, more endoscopists need to receive training on this procedure. Currently, there are not a lot of recommendations concerning this topic published in the literature reflecting lack of high evidence level studies.

## Figures and Tables

**Figure 1 medsci-12-00009-f001:**
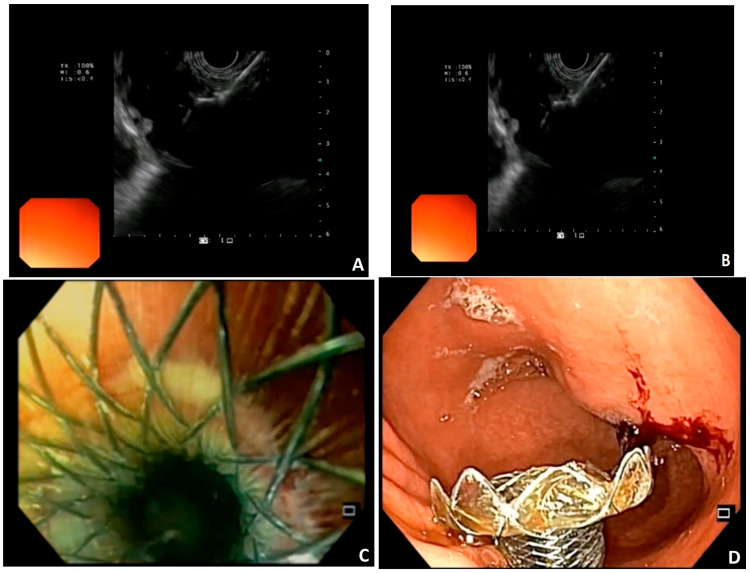
Images courtesy of Dr. Benedetto Mangiavillano. EUS-guided Gastroenterostomy. (**A**,**B**) placement of LAMS in the jejunal loop; (**C**,**D**) endoscopic view of the gastrojejunostomy.

**Table 1 medsci-12-00009-t001:** Advantages and disadvantages of techniques employed for managing gastric outlet obstruction.

Method	Advantages	Disadvantages
Surgical Gastroenterostomy	Long-term durability Salvage solution if endoscopic treatments have failed	Invasive method High morbidity, contraindicated in critically ill patients Gastroparesis
SEMS	Less invasive, safe Widely available in daily clinical practice Rapid alleviation of symptoms, early resumption of chemotherapy, oral intake	High reintervention rate due to stent obstruction
EUS-GE	Less invasive, safe procedure compared to SGE Sustained patency, long-term efficacy Rapid alleviation of symptoms, early resumption of chemotherapy, oral intake Feasible in patients with concomitant biliary obstruction	Not standardized More expertise is required Poor performance in case of uncontrolled ascites, diffuse peritoneal disease, or diffuse infiltration of gastric wall

## Data Availability

Not applicable.

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
