# Peer review of "Revealing Insights: A Comprehensive Overview of Gastric Outlet Obstruction Management, with Special Emphasis on EUS-Guided Gastroenterostomy"

_medsci, 2024, doi:10.3390/medsci12010009_

Round 1

Reviewer 1 Report

Comments and Suggestions for Authors

Dear Editor,

I read with great interest this Review on GOO and EUS-GE.

The review covers this new, fascinating, topic extensively, detailing evidence, advantages, pitfalls of different techniques, and I think it might be useful for the readers.

However, I found some issues that should be addressed before the acceptance of the paper.

Major

1. The Authors use extensive lists of events when describing events. As for example, when dealing with causes of benign GOO, they cite almost 20 diagnoses. This is epidemiologically irrelevant. The same happens when they describe complications of SGE, without critically appraise what are the most frequent and relevant (I cannot imagine nutritional deficiency due to long bypassed intestine it the surgeon knows what he is doing, whereas infections or delayed gastric emptying is much more frequent due to peristalsis perturbation in open surgeries).

2.  When describing techniques, the Authors have performed a detailed analysis of several methods. However, they did not describe

a.  one of the most frequently used techniques, the so-called WEST (The wireless EUS-guided gastroenterostomy simplified technique). This corresponds to the “Antegrade direct” methods described by the Authors, but without the need for the 19G needle puncture.

b. A technique which is especially useful when a guidewire cannot pass the stricture, the so-called nonassisted direct technique over a guidewire (DTOG).

Moreover, the Authors describe all techniques as they are identical, whilst some techniques are very rarely used (as for example the retrograde).

A recent study has compared the WEST and DTOG techniques (10.1055/a-2119-7529)

3. I strongly suggest to remove reference 21, or at least amend what the Authors extracted from this reference. The Authors tell “Wannhof et al. aimed to determine factors affecting technical success in 35 patients under-209 went EUS-GE for GOO. On multivariate analysis, the distance between the two lumina where LAMS was deployed was the only predictor of technical success, with a cut-off value of 19mm as the best distance”. First of all, the Authors reported a wrong cut-off, because this was 10mm (according to Youden’s index) in the Manuscript. Moreover, this cut-off was determined on preprocedural CT, which is completely useless, as the relation of the jejunum to the stomach should be assessed after distention of the loop with fluids. I strongly discourage to cite this evidence. The distance between the two lumens should be in the order of a few millimeters to avoid complications.

4.  In the comparison of EUS-GE versus SEMS, the Authors do not cite recent relevant prospective evidence, the randomized study by Teoh AYB et al (10.1016/S2468-1253(23)00242-X) and the prospective comparison by Vanella et al (10.1016/j.gie.2023.04.2072)

5.  When dealing with EUS double bypass the most recent evidence is Bronswijk et al (10.1016/j.gie.2023.03.019)

Minor

1.  When dealing with diagnostic flow-chart of GOO, the authors focus on CT and gastroscopy, and do not cite Radiological Contrast Follow-through. This diagnostic modality is especially useful in patients with known malignancy and new onset of vomiting. Conversely, I’ve rarely performed an upper GI endoscopy for GOO, as in most cases these patients are known oncological patients.

2.  Introduction: SGE, SEMS and EUS-GE have never been spelled before (only in the abstract)

3.  “More synchronous” endoscopic procedures is misleading. Maybe the Authors meant more modern / less invasive?

4.  When describing LAMS, I would avoid details alluding to specific LAMS designs (“flared ends of 24mm for the 15mm stent”)

5.  "duodenal tube” is misleading. Please refer to an orojejunal tube.

6.  I would not cite balloon dilation as a salvage method of type 1 misdeployment

7.  NOTES is not “placed”. NOTES refers to a group of techniques for completing the procedure working into the peritoneum

Comments on the Quality of English Language

Ok

Author Response

Dear Reviewer,

Major revision 1

The Authors use extensive lists of events when describing events. As for example, when dealing with causes of benign GOO, they cite almost 20 diagnoses. This is epidemiologically irrelevant. The same happens when they describe complications of SGE, without critically appraise what are the most frequent and relevant (I cannot imagine nutritional deficiency due to long bypassed intestine it the surgeon knows what he is doing, whereas infections or delayed gastric emptying is much more frequent due to peristalsis perturbation in open surgeries).

Response:

Thank you for pointing this out. It is evident that the diversity of examples concerning events is inconsequential and has the potential to alter the mean of the more typical instances.

Consequently, we have revised the corresponding sections of the manuscript, and we hereby present the updated content:

<<Other benign conditions causing GOO are post ulcer stenosis, acute pancreatitis and pancreatic fluid collections, radiation-induced stricture, caustic ingestion, foreign body, benign tumor (adenoma, lipoma), Crohn´s disease and eosinophilic gastroenteritis>>.

<<Common complications of the SGE include infections, gastroparesis, hemorrhage, anastomotic leak (mostly between the third to fifth postoperative day) and  marginal ulcer (gastrojejunostomy predisposes the part of the jejunum nearest to the stomach is susceptible to ulcer development because it lacks the protective mechanisms of the duodenum)>>.

Major revision 2

 When describing techniques, the Authors have performed a detailed analysis of several methods. However, they did not describe

  1. one of the most frequently used techniques, the so-called WEST (The wireless EUS-guided gastroenterostomy simplified technique). This corresponds to the “Antegrade direct” methods described by the Authors, but without the need for the 19G needle puncture.
  2. A technique which is especially useful when a guidewire cannot pass the stricture, the so-called nonassisted direct technique over a guidewire (DTOG).

Moreover, the Authors describe all techniques as they are identical, whilst some techniques are very rarely used (as for example the retrograde).

A recent study has compared the WEST and DTOG techniques

Response:          

In response to your recommendations, we have successfully incorporated both the WEST and DTOG techniques, providing comprehensive descriptions and ensuring a thorough comparison through the addition of relevant references.

Additionally, we have organized the EUS-GE techniques into categories based on their frequency of use, a modification aimed at improving the clarity and coherence of our analysis.

‘’2.Antegrade EUS-GE Direct method using the wireless endoscopic simplified technique (WEST)

Step 1: as in Technique 1.

Step 2: Advance and deploy directly a cautery-enhanced LAMS (HOT AXIOS; Boston Scientific Corp.) in cut mode.

4.Antegrade EUS-GE with direct technique over a guidewire (DTOG)

This technique is applied, if the obstruction cannot be crossed.

Step 1: Administer intravenous anticholinergic agent to slow bowel movements

Step 2: Puncture the target intestinal limb with a 19-gauge needle.

Step 3: Fill the jejunal limb with contrast medium through the needle and insert a guidewire using fluoroscopy.

Step 4: Advance the cautery-enhanced LAMS catheter over the guidewire into the je-junal limb in cut mode. Deploy the LAMS.’’

Major revision 3

I strongly suggest to remove reference 21, or at least amend what the Authors extracted from this reference. The Authors tell “Wannhof et al. aimed to determine factors affecting technical success in 35 patients under-209 went EUS-GE for GOO. On multivariate analysis, the distance between the two lumina where LAMS was deployed was the only predictor of technical success, with a cut-off value of 19mm as the best distance”. First of all, the Authors reported a wrong cut-off, because this was 10mm (according to Youden’s index) in the Manuscript. Moreover, this cut-off was determined on preprocedural CT, which is completely useless, as the relation of the jejunum to the stomach should be assessed after distention of the loop with fluids. I strongly discourage to cite this evidence. The distance between the two lumens should be in the order of a few millimeters to avoid complications.

Response:

We would like to express our gratitude for bringing attention to the issue of the distance between the two lumina before anastomosis.

We acknowledge your concern regarding the variation in the cut-off values for the distance between the two lumina before anastomosis in this study. The initial cut-off of 10 mm, subsequently adjusted to 19 mm after excluding the first 13 cases, was indeed a deliberate adaptation by the authors. This modification aimed at mitigating the impact of the learning curve.

Additionally, your point about addressing the distance between the two lumina while filled with water, reflecting the real situation during the procedure, is well-taken.

In light of these considerations, we have decided to remove the section related to this aspect from the article. We believe this modification will strengthen the overall integrity of our paper.

Major revision 4

In the comparison of EUS-GE versus SEMS, the Authors do not cite recent relevant prospective evidence, the randomized study by Teoh AYB et al (10.1016/S2468-1253(23)00242-X) and the prospective comparison by Vanella et al.

Response:

We concur that the inclusion of both prospective and randomized data is of utmost relevance to the comprehensiveness of our study.

Regarding the study by Teoh et al., which represents the first randomized trial comparing EUS-GE with other therapeutic modalities, we were aware of its upcoming publication. Unfortunately, up until the final stages of preparing the manuscript, the anticipated publication had not occurred.

The exposition of data from the aforementioned studies is situated within the section titled "EUS-GE vs SEMS" in our manuscript, as delineated below:

‘’Similarly, a recent prospective study conducted at a single center investigated the efficiency of EUS-GE via the WEST method, vs SEMS with 28 patients undergoing EUS-GE and 28 matched patients undergoing SEMS. The EUS-GE group demonstrated superior clinical success (100% vs 75.0%, P = .006) and a marked decrease in symptom relapse throughout the observation period (3.7% vs 33.3%, P = .02). Furthermore, EUS-GE was associated with a reduced time to clinical success vs SEMS (2 vs 3 days, P = .03). [10.1016/j.gie.2023.04.2072]. Teoh AYB et al. [10.1016/S2468-1253(23)00242-X] reported the first multicenter, randomized, controlled trial assessing EUS-GE, by using the EPASS technique (n=48), vs SEMS (n=49) for the treatment of malignant GOO. Regarding the primary outcome of reintervention rate at 6 months, EUS-GE showed notable superiority, with only two patients (4%) requiring reintervention compared to 14 patients (29%) in the SEMS group (p=0.0020; RR 0.15 [95% CI 0.04–0.61]). EUS-GE was also linked with longer stent patency and improved eating function at 1 month, as reflected by a significantly higher mean 1-month Gastrointestinal Quality of Life Index (GOOS) of 2.41 (SD 0.7) compared to the SEMS group with a mean of 1.9 (0.9; p=0.012, r=0.26). Technical success, clinical success and adverse effects were comparable between the two interventions.’’

Major revision 5

When dealing with EUS double bypass the most recent evidence is Bronswijk et al

Response:

In response to your suggestion, we have made a modification to the specific section. We added the most recent and pertinent data, thereby ensuring the accuracy and currency of the information presented.

Minor revision 1

When dealing with diagnostic flow-chart of GOO, the authors focus on CT and gastroscopy, and do not cite Radiological Contrast Follow-through. This diagnostic modality is especially useful in patients with known malignancy and new onset of vomiting. Conversely, I’ve rarely performed an upper GI endoscopy for GOO, as in most cases these patients are known oncological patients.

Response:

We  express our appreciation for your insightful comments concerning the role of contrast studies in the diagnostic flowchart of gastric outlet obstruction.

After careful consideration, we have chosen to incorporate your feedback and have accordingly revised this section of our manuscript :

 ‘’Contrast studies utilizing barium or water-soluble contrast agents prove to be valuable diagnostic tools, particularly in patients with a documented history of malignancy who present with a sudden onset of vomiting. The absence of contrast passage in the small intestine indicating a complete GOO’’.

Minor revision 2

Introduction: SGE, SEMS and EUS-GE have never been spelled before (only in the abstract)

We have implemented the necessary modifications as per your recommendations.

Minor revision 3

“More synchronous” endoscopic procedures is misleading. Maybe the Authors meant more modern / less invasive?

We have implemented the necessary modifications as per your recommendations.

‘’ the emergence of less invasive endoscopic techniques.’’

Minor revision 4

When describing LAMS, I would avoid details alluding to specific LAMS designs (“flared ends of 24mm for the 15mm stent”)

We have implemented the necessary modifications as per your recommendations.

Minor revision 5

"duodenal tube” is misleading. Please refer to an orojejunal tube.

We have implemented the necessary modifications as per your recommendations.

‘’ through an orojejunal tube’’.

Minor revision 6

I would not cite balloon dilation as a salvage method of type 1 misdeployment

We have implemented the necessary modifications as per your recommendations.

‘’ This Type of error was addressed during the same endoscopic session via EUS-GE at the same or different gastric site or via placement of a duodenal stent .’’

Minor revision 7

NOTES is not “placed”. NOTES refers to a group of techniques for completing the procedure working into the peritoneum

We have implemented the necessary modifications as per your recommendations.

You will find these segments also in the revised manuscript.

Thank you again for your support.

Best regards,

Dimitrios Ziogas

Reviewer 2 Report

Comments and Suggestions for Authors

This paper provides a detailed summary of endoscopic and surgical approaches to GOO. In particular, the endoscopic ultrasound approach is a new procedure, has a high incidence of complications, and requires more time to standardize the procedure. It would be better to provide readers with an easy-to-understand schema for each technique.

Author Response

Dear Reviewer,

I would like to express my sincere gratitude for dedicating your time to thoroughly review our manuscript and for providing valuable comments.

 In response to your suggestion to enhance the clarity of the techniques, we have implemented more detailed descriptions in each section of the manuscript. These revisions aim to make the content more accessible and comprehensible for our readers.

You will find these segments also in the revised manuscript.

Thank you again for your support.

Best regards,

Dimitrios Ziogas

Round 2

Reviewer 1 Report

Comments and Suggestions for Authors

Dears Editors, 

The authors have replied to all raised concerns, and I'm satisfied with the revisions, improving the quality of the Manuscript.